# Evaluation of Liver Stiffness Measurement by Means of 2D-SWE for the Diagnosis of Esophageal Varices

**DOI:** 10.3390/diagnostics13030356

**Published:** 2023-01-18

**Authors:** Bozhidar Hristov, Vladimir Andonov, Daniel Doykov, Katya Doykova, Siyana Valova, Emiliya Nacheva-Georgieva, Petar Uchikov, Gancho Kostov, Mladen Doykov, Eduard Tilkian

**Affiliations:** 1Second Department of Internal Diseases, Section “Gastroenterology”, Medical Faculty, Medical University of Plovdiv, 6000 Plovdiv, Bulgaria; 2Gastroenterology Clinic, University Hospital “Kaspela”, 4001 Plovdiv, Bulgaria; 3Department of Diagnostic Imaging, Medical Faculty, Medical University of Plovdiv, 6000 Plovdiv, Bulgaria; 4Department of Diagnostic Imaging, University Hospital “Kaspela”, 4001 Plovdiv, Bulgaria; 5Second Department of Internal Diseases, Section “Nephrology”, Medical Faculty, Medical University of Plovdiv, 6000 Plovdiv, Bulgaria; 6Clinic of Nephrology, University Hospital “Kaspela”, 4001 Plovdiv, Bulgaria; 7Department of Special Surgery, Faculty of Medicine, Medical University of Plovdiv, 6000 Plovdiv, Bulgaria; 8St. George University Hospital, 4000 Plovdiv, Bulgaria; 9Department of Surgery, University Hospital “Kaspela”, 4001 Plovdiv, Bulgaria; 10Department of Urology and General Medicine, Medical Faculty, Medical University of Plovdiv, 6000 Plovdiv, Bulgaria; 11Clinic of Urology, University Hospital “Kaspela”, 4001 Plovdiv, Bulgaria

**Keywords:** 2D-SWE, elastography, portal hypertension, liver cirrhosis, esophageal varices

## Abstract

Portal hypertension (PH) and esophageal varices (EVs) are a matter of extensive research. According to current Baveno VII guidelines, in patients with compensated advanced chronic liver disease (cACLD), liver stiffness measurement (LSM) < 15 kPa and PLT count > 150 × 10^9^/L, upper endoscopy (UE) is not mandatory, and the emphasis should be set on non-invasive methods for evaluation of clinically significant portal hypertension (CSPH). The aim of this study is to establish whether liver stiffness (LS) measured by 2D-SWE could be used as a predictor for the presence and severity of EVs in cirrhotic patients. In total, 86 patients of whom 32 with compensated liver cirrhosis (cLC) and 54 with decompensated liver cirrhosis (dLC) were examined in the Gastroenterology clinic of University hospital “Kaspela”, Plovdiv, Bulgaria. Each patient underwent LS assessment by 2D-SWE and EVs grading by UE. EVs were detected in 47 (54.7%) patients, 23 (49%) of them were stage 4-high-risk EVs (HREV). The cut-off value for LS that differentiates HREV from the rest was set at 2.49 m/s with 100% sensitivity and 100% specificity (AUC 1.000, CI 0.925). Conclusions: 2D-SWE can be used as a non-invasive method in the assessment of only high-grade esophageal varices. For the other grades, upper endoscopy remains the method of choice.

## 1. Introduction

Portal hypertension (PH) is a major consequence of liver cirrhosis (LC) associated with its most grave complications including bleeding from EVs, ascites, and hepatic encephalopathy. Research shows that the presence of CSPH (defined as HVPG > 10 mmHg) is a strong predictor for death in patients with cACLD [1]. Traditionally the “gold standard” for the estimation of PH has been the measurement of the hepatic venous pressure gradient (HVPG) [2]. Though considered invaluable in clinical research settings, its usage in real clinical practice is generally narrowed by its invasiveness [2]. For the establishment of CSPH, the most extensively utilized method is diagnostic UE to screen for the presence and severity of EVs. According to recent Baveno VII consensus guidelines on PH, in patients with cACLD, the emphasis should be set on the non-invasive evaluation of PH. It is postulated that the presence of CSPH is highly unlikely in patients with LSM < 15 kPa (measured through Transient elastography (TE)) and PLT > 150 × 10^9/^L (sensitivity and negative predictive value (NPV) > 90%), therefore such patients could be followed annually by non-invasive techniques [2]. In patients unfit for treatment with non-selective beta-blockers (NSBB) screening endoscopy is not mandatory as well, if LSM is <20 kPa and PLT > 150 × 10^9/^L [2]. Of note, the etiology of liver damage is also a factor, the guidelines being applicable mainly in viral, alcoholic liver disease, and non-alcoholic steatohepatitis (NASH) with BMI < 30 kg/m^2^. Splenic stiffness measurement (SSM) is also used as a surrogate marker for PH, with cut-off values for ruling out and in CSPH being <21 kPa and >50 kPa respectively.

As already outlined, the main diagnostic modality to evaluate tissue stiffness is ultrasound elastography. Elasticity is the physical feature of substances to regain their original state after the force that caused their deformation has stopped [3]. Recently, several elastographic methods for non-invasive assessment of liver fibrosis have been studied: transient elastography, point shear-wave elastography (pSWE), two-dimensional shear wave elastography (2D-SWE), and magnetic resonance elastography (MRE). In most recent Baveno VII guidelines the chief diagnostic modality utilized is TE. The guideline highlights the need to establish reference values for LS in other elastographic techniques, namely pSWE and 2D-SWE. The 2D-SWE technique has numerous theoretical advantages over other elastographic techniques particularly the large size of the explored area, the ability to simultaneously visualize the region of interest (ROI) in B-mode and to monitor and evaluate the quality of the measurements in real-time [3,4]. Nevertheless, 2D-SWE has not yet been extensively used in clinical routine, and data regarding its applicability for the evaluation of CSPH and EVs are scarce [4].

In the current study, we evaluated the diagnostic potential of 2D-SWE for the non-invasive diagnosis of EVs. As a secondary endpoint, we studied the correlation of shear wave velocity values (SWV) with various factors including sex, presence of ascites, and liver enzymes. Eventually, cut-off values for the diagnosis of EVs and HREVs were estimated.

## 2. Materials and Methods

Study design: Prospective single-center comparative study. A total of 86 patients from among the patients admitted to University Hospital “Kaspela” for diagnosis and/or treatment of liver cirrhosis between January and November 2021 were included. Subjects were divided into two groups as follows: (1). Compensated liver cirrhosis, (2). Decompensated liver cirrhosis. The presence of either ascites, prior episode of variceal hemorrhage, hepatic encephalopathy, or jaundice were considered as criteria for inclusion in the dLC group, while the rest were selected for the cLC group. Patients with thrombosis of the portal venous system, prior endoscopic band ligation (EBL), obstructive jaundice, or verified malignant lesions in the liver we considered ineligible and were excluded from the analysis.

Each of the participants underwent a physical examination and biochemical testing including alanine aminotransferase (ALAT), aspartate aminotransferase (ASAT), alkaline phosphatase, gamma-glutamyl transferase, bilirubin, and platelet count. Each participant underwent conventional abdominal ultrasound in B-mode to assess the size, shape, and structure of the liver. Standard liver measurements were performed: the ventro-dorsal size of the liver (mm) on the right midclavicular line, portal vein size (mm), and portal blood flow velocity by Doppler US (m/sec). The presence of ascites and visible portosystemic anastomoses was also recorded.

LSM was performed by 2D-SWE using the QElaXto^®^ software, version F104513Q101701 of an Esaote MyLab™ 9 eXP (Genoa, Italy) ultrasound device. A convex C1-8IQ appleprobe transducer was used. Ten elastographic measurements were performed with a validation criterion-median interquartile range (IQR/M) <30% in accordance with current recommendations [5].

LS was assessed elastographically by examining the right hepatic lobe of each patient through the intercostal space. Patients were in a supine position with their right arm at maximum abduction in compliance with the latest requirements for quality measurement [6,7]. During standard B-mode, an assessment area was selected with no large blood vessels. Location at a distance of at least 1.5 cm from the Glisson’s capsule was selected and the ROI size was fixed by the device. The patient was instructed to hold his breath while the values were calculated (Figure 1). If the colour box was not filled by > 50% of its surface or if breathing was uncontrolled, the elastogram was discarded and a new acquisition was attempted. Ten LSMs were obtained, and the median value in m/s was used for analysis.

Upper endoscopy was utilized as a reference method to evaluate the presence and grade of EVs. An endoscopic processor Olympus Evis Exera III (Hamburg, Germany), coupled with the Olympus GIF-HQ190 (Hamburg Germany) therapeutic gastroscope, was used. Grading of EVs was performed in concordance with the Paquet classification (Table 1) [8]. Moderate insufflation was used. All procedures were performed by a single endoscopist to ensure adequate comparability.

## 3. Results

A total of 86 patients were examined and stratified into two groups in accordance with the outlined criteria: 32 with cLC and 54 with dLC. Demographic and lab data of the patients are presented in Table 2

The average age of the studied population was 63 ± 8.03 years with an age range between 48 and 81 years. Males were significantly prevalent with a relative share of 60.5% (N = 52) compared to women–39.5% (N = 34), *p* = 0.009. The sex distribution is similar in the two groups of patients: compensated cirrhosis—59.40% men; decompensated cirrhosis — 61% men, with no significant difference (*p* = 0.874). The average age of patients with compensated cirrhosis (65.50 ± 9.13 years) was higher than that of patients with decompensated cirrhosis (61.54 ± 6.97 years), *p* = 0.039.

Body mass index (BMI) for the whole group had an average value of 28.78 ± 2.69 kg/m^2^ with a minimum value of 22.60 kg/m^2^ and a maximum value of 35.30 kg/m^2^. In patients with decompensated cirrhosis, significantly higher BMI values were established (29.36 ± 2.64 kg/m^2^) compared to patients with compensated cirrhosis (27.80 ± 2.51 kg/m^2^), *p* = 0.009.

The biochemical tests ASAT, ALAT, and AP showed similar values in the whole group and in both subgroups with no significant difference (*p* > 0.05 for the three indicators). Significantly higher GGT values were observed in patients with compensated cirrhosis (136.22 ± 94.10 IU/L) compared to those with decompensated cirrhosis (87.80 ± 48.67 IU/L), *p* = 0.01. Of note though, in both groups, the values were much higher than the upper reference limit (30 IU/L).

UE established EVs in 47 of 86 patients (54.65%). Based on Paquet classification patients with EVs were distributed as follows: first grade—9 patients, second grade—9 patients, third grade—6 patients, and fourth grade—23 patients.

As anticipated in patients with dLC, consistently higher mean SWV values were found–2.41 ± 0.31 m/s compared to 2.24 ± 0.13 m/s for patients with cLC, *p* = 0.001.

With a cut-off value of 2.18m/s 2D-SWE showing an accuracy of 71.80% for the identification of EVs (irrespective of grade) (AUC = 0.718, *p* < 0.001), with 68% sensitivity and 57% specificity. The second analysis, investigating the diagnostic value of 2D-SWE for the distinction between HREVs (grade 4) and the other three grades and patients with no EVs showed 100% accuracy (AUC = 1000, *p* < 0.001). The cut-off value for the diagnosis of grade 4 varices was set at 2.49 m/s with 100% sensitivity and 100% specificity (Table 3). Of note, the diagnostic accuracy of 2D-SWE was comparable between the investigated groups (*p* = 0.360), which justified the calculation of a single cut-off value for both groups.

ROC curves depicting the performance of 2D-SWE for the diagnosis of EVs and HREVs are presented on Figure 2.

As a secondary endpoint, the correlation of SWV values measured through 2D-SWE with certain factors was estimated. In both studied groups, sex was found to have no significant effect on SWV values. Men showed a higher average value (2.39 ± 0.30 m/s) than women (2.29 ± 0.22 m/s) but with a minimum difference of 0.10 m/s and no statistical significance, *p* = 0.08.

A similar comparative analysis was performed between patients with ascites (N = 27) and those without ascites (N = 59). In the group with ascites, a significantly higher value of SWV (2.48 ± 0.31 m/s) was found in comparison with the group without ascites (2.28 ± 0.23 m/s), *p* = 0.006.

SWV showed a high positive correlation with lab parameters ASAT (r = 0.745, *p* < 0.001), ALAT (r = 0.753, *p* < 0.001), and a lower one with AP (r = 0.358, *p* < 0.001) (Figure 3).

## 4. Discussion

Non-invasive methods for the assessment of liver fibrosis and its complications have been subject to numerous studies in recent years. Cirrhotic transformation of the parenchyma regardless of its etiology is associated with structural and biochemical changes resulting in increased portal pressure [9]. Those changes are related to alterations in vascular architecture, parenchymal fibrosis, and impaired clearance of endogenic vasodilators [10].

The relevance of CSPH was already outlined, with its progression being positively associated with mortality rate [11]. Classically, HVPG is considered the gold standard for the diagnosis of PH, as UE is for EVs, one of the main clinical manifestations of PH. Unfortunately, measurement of HVPG proves to be quite invasive and hardly applied outside clinical research settings, while the necessity for regular UE to monitor EVs, makes the procedure inconvenient and not cost-effective. Therefore, the development of non-endoscopic and non-invasive methods for the assessment of EVs is important both for periodic follow-ups well as for the identification of the candidates for interventional treatment [12].

Numerous authors conduct studies to evaluate esophageal varices by noninvasive methods, including assessment of liver stiffness by various elastographic techniques, platelet count, platelet count/spleen size, or spleen stiffness [13,14,15,16].

Grgurevic et al. [17] in a study of 87 patients with cirrhosis (predominantly chronic viral hepatitis C and alcoholic cirrhosis), established EVs in 54 of them. The criterion value for the presence of EVs was 19.7 kPa with a sensitivity of 83.3% and a specificity of 66.6%. Those findings are in conflict with our study which found that 2D-SWE performs suboptimally in identifying patients with low-grade EVs. With a cut-off value of 2.18m/s, the diagnostic accuracy of LSM was 71.8% with sensitivity and specificity of merely 57% and 68% respectively. In spite of those disappointing results, further assessment established that 2D-SWE has excellent performance (sensitivity and specificity of 100%) regarding the diagnosis of high-grade varices (high-risk esophageal varices). This finding is supported by a recent meta-analysis which found 80% sensitivity and 73.1% specificity of 2D-SWE for the diagnosis of HREVs [18]. Research by Arun et al. finds that 2D-SWE has good diagnostic capabilities for all grades of EVs including HREVs in patients with cACLD [19]. They also note that the accuracy of 2D-SWE is comparable to TE but superior to all other non-invasive methods.

It is our opinion that the discrepancy between our results and those cited in the literature is mainly derived from the heterogeneity of the population in our study including both patients with compensated and decompensated liver cirrhosis. Our data shows that the stage of liver cirrhosis has little influence on the predictive value of 2D-SWE for the diagnosis of HREV. This would suggest that 2D-SWE may be successfully utilized for the identification of patients with dLC and HREVs, who would benefit most from the performance of UE, as it would be both diagnostic and therapeutic. Such a hypothesis urges further research.

In current literature, the cut-off value for the presence of EVs, measured through pSWE, was set at 2.05 m/s with good sensitivity (83%), specificity (76%), positive predictive value (PPV) (78%), negative predictive value (NPV) (81%). For HREVs the cutt-off value is 2.39 m/s with sensitivity–81%, specificity–82%, PPV–69%, and NPV–89% [20]. Although understandably not entirely applicable. Those results are largely similar to the ones obtained in our study.

When analyzing the acquired data, certain findings were encountered. Patients with decompensated cirrhosis were generally younger compared to those with compensated disease (*p* = 0.039). Such observations might be at least partially explained by the gradual decrease of certain complications such as bleeding in the course of the disease [21]. In subjects with decompensated liver disease consistently higher values of BMI and GGT levels were established. In our opinion, this finding reflects the tendency towards shifting the etiology of the disease from mainly viral to alcoholic and metabolic. This statement is supported by a large study that established a more than two-fold increase in obesity in cirrhotic patients between 2008 and 2018. In 2018 BMI > 40 kg/m^2^ is measured in 27% of cirrhotic patients [21].

The correlation of SWV values measured through 2D-SWE with certain factors was also investigated. In patients with ascites consistently higher SWV values were established.

Since both SWV and ascites are closely dependent on the severity of portal hypertension such a finding was largely anticipated. Transaminase levels also showed a positive correlation with SWV. This observation is repeatedly confirmed in other studies, but its clinical significance is yet to be determined [22].

The presented research has a few weaknesses. First, it reviews patients with both cLC and dLC, which reflects on the final results. Secondly obtained data is presented in m/s, while most studies use kPa as a metric. Our decision was driven by the recent EFSUMB and WFUMB guidelines which regard tissue stiffness measurement in m/s to be more objective and precise [6,7]. Certain issues with comparability cannot be denied. On the other hand, we think that the inclusion of patients irrespective of stages of LC would expand the knowledge and raise the need for future research on 2D-SWE in patients with dLC. Our initial findings suggest that 2D-SWE may be a valuable tool for the identification of HREVs in subjects with dLC.

## 5. Conclusions

The 2D-SWE technique can be used as a non-invasive method in the diagnostic algorithm for patients with both compensated and decompensated liver cirrhosis to assess the presence of high-risk esophageal varices. It could be used as a tool to identify patients who would benefit the most from UE (UE with intent to treat). In fact, 2D-SWE might be valuable for the establishment of low-grade EVs as well, but future research is needed to support such a statement.

## Figures and Tables

**Figure 1 diagnostics-13-00356-f001:**
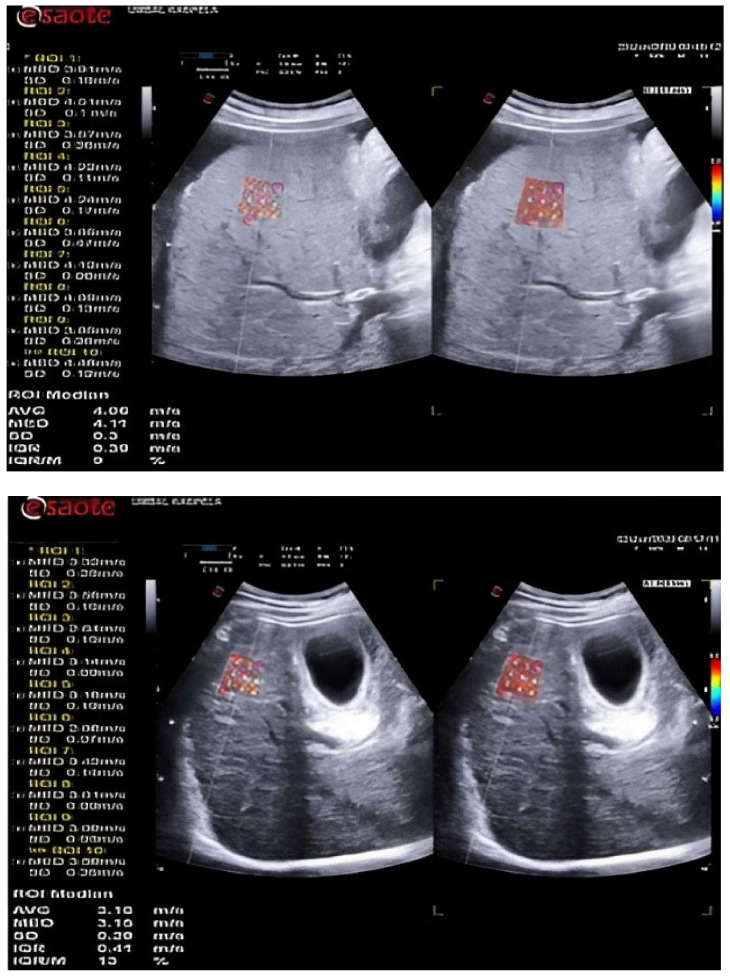
2D-SWE images in a patient with liver cirrhosis.

**Figure 2 diagnostics-13-00356-f002:**
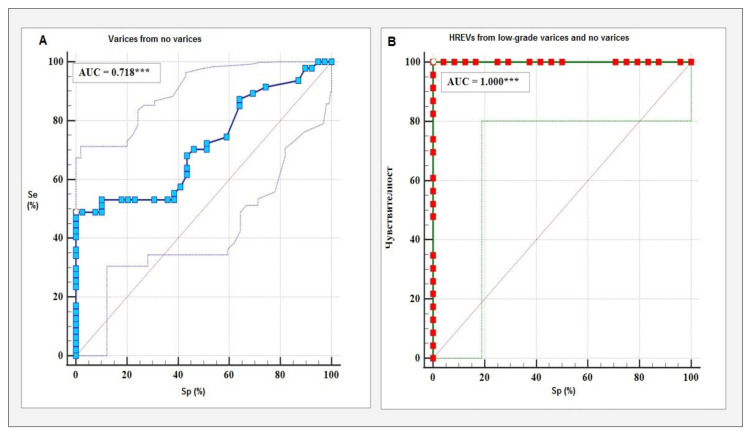
ROC curves of diagnostic accuracy of 2D-SWE; (**A**) Differentiation of patients with EVs from those with no varices, (**B**) Differentiation of grade 4 varices from no varices and EVs grades 1, 2, and 3. Se—sensititity; Sp—specificity; ***—Statistical significance at *p* ≤ 0.001.

**Figure 3 diagnostics-13-00356-f003:**
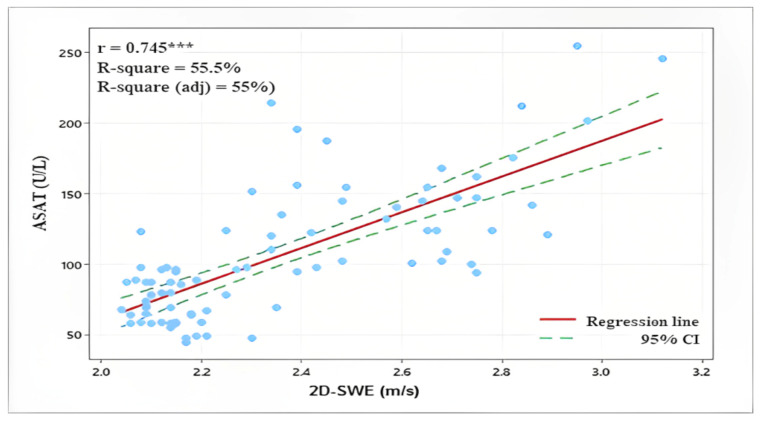
Correlation between liver 2D-SWE and biochemical blood indicators ASAT, ALAT, and alkaline phosphatase. ***—Significant assossiation (*p* < 0.001); r—Pearson’s correlation coefficient; R-suqare—coefficient of determination.

**Table 1 diagnostics-13-00356-t001:** Paquet classification of EVs.

Grade	Endoscopic Appearance
I	Varices extending just above the mucosal level
II	Varices projecting by one-third of the luminal diameter that cannot be compressed with air insufflation
III	Varices projecting up to 50% of the luminal diameter and in contact with each other
IV	As III + presence of red spots or strips or small varices lying on larger ones (varices on varices)

Data analysis was performed using the statistical program IBM SPSS, version 26 (2018), the specialized program for medical analysis MedCalc version 19.4.1 (2020), and the statistical program Minitab version 19 (2020). Kolmogorov-Smirnov test, independent-samples *t*-test, Fisher’s exact test, and Pearson r-correlation were used. To establish the potential of LSM for the diagnosis of esophageal varices in cirrhotic patients a receiver operating characteristic (ROC) curve was determined, the area under the ROC (AUROC) was evaluated and the optimal cut-off values for the diagnosis of EVs and particularly HREV were calculated.

**Table 2 diagnostics-13-00356-t002:** General demographic and lab data.

Variables	Total (N = 86)	Group		
		Compensated Cirrhosis (N = 32)	Decompensated Cirrhosis (N = 54)	*p*
Age $\bar{X}\;$± SD	63 ± 8.03	65.50 ± 9.13	61.54 ± 6.97	0.039 *
Sex N (%) male female	52(60.50%) ** 34 (39.50%)	19 (59.40%) 13(40.60%)	33 (61%) 21 (39%)	0.874
BMI kg/m^2^ $\bar{X}$ ± SD	28.78 ± 2.69	27.80 ± 2.51	29.36 ± 2.64	0.009 **
ASAT U/L $\bar{X}$ ± SD	105.27 ± 46.90	102.13 ± 45.43	107.13 ± 48.08	0.635
ALAT IU/L $\bar{X}$ ± SD	122.87 ± 46.78	121.19 ± 44.16	123.87 ± 48.65	0.631
GGT IU/L $\bar{X}$ ± SD	105.81 ± 72.53	136.22 ± 94.10	87.80 ± 48.67	0.01 **
AP IU/L $\bar{X}$ ± SD	92.13 ± 46.81	81.59 ± 41.13	98.17 ± 42.44	0.121
Ascites N (%)	27 (31.40%)	3(9.40%)	24 (44.4%)	0.001 ***
Varices N (%) Grade 1(А) Grade 2(В) Grade 3(С) Grade 4(D)	47 (54.70%) 9 (19.00%) 9 (19.00%) 6 (13.00%) 23 (49.00%)	0 (0%)	47 (87%)	0.000 ***

X ®= arithmetic mean value; N—Number of patients; ASAT-Aspartate aminotransferase; ALAT-Alanine transaminase; GGT-gamma-glutamyl transferase; AP-; *–Statistical significance at *p* < 0.05; **—Statistical significance at *p* ≤ 0.01; ***—Statistical significance at *p* ≤ 0.001.

**Table 3 diagnostics-13-00356-t003:** Cut-off values for the diagnosis of EVs and HREVs.

Statistics	Presence of Varices from Absence	Varices Grade 4 from Grades 1,2,3 and no Varices
AUC	0.718	1.000
95% CI	0.610 to 0.810	0.925 to 1.000
SE	0.055	0.000
p	0.000 ***	0.000 ***
Cut-off value	> 2.18m/s	> 2.49m/s
Sensitivity	68%	100%
Specificity	57%	100%

AUC—area under the curve; SE—standard error; ***—Statistical significance at *p* ≤ 0.001.

## Data Availability

The data presented in this study are available in article.

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
