# Peer review of "Evaluation of Liver Stiffness Measurement by Means of 2D-SWE for the Diagnosis of Esophageal Varices"

_diagnostics, 2023, doi:10.3390/diagnostics13030356_

Round 1

Reviewer 1 Report

This manuscript explores the potential to diagnose the esophageal varices using liver stiffness as the biomarker. Many patients were involved in this study which could lead to some interesting findings. However, this manuscript is not well-written, it is lack logistic relations between sentences and paragraphs. In the Result part, it is not well presented as well, I can not tell the relation between the SWE and other methods. In the Discussion part, the results are expected to have more descriptions. Overall, the clinical study is interesting, but the manuscript needs extensive editing.

Minor comments: English should be improved. The abbreviations are not regulatory. 

Reviewer 2 Report

Dear Authors,

In my opinion, the reviewed article is correctly written and very interesting. It might be of great interest for hepatologists or clinicians who perform liver elastography. The English language used in the article is generally correct.

For me, issues to be clarified or improved are as follows:

1.  line 145 - "p = 0.009" - it seems to me that the statement about prevalence of males compared to women has no coverage in the data in table 2 ???

2.  lines 147-159 - You describe many interesting results: 1. more younger patients with decompensated cirrhosis than older pts -> why - alcohol consumption, more concomitant diseases, others ? ... 2. more patients with decompensated cirrhosis & significantly higher BMI -> why - ascites, metabolic problems, others ? ... 3. significantly lower GGT levels in patients with decompensated cirrhosis -> why - less alcoholics, due to chronic liver insufficiency ? ... comments, comments, comments !!! - I miss them so much in the end of the article, in discussion section :) ... 

3. lines 248-263 - Congratulations for honest self-criticism on the results, it's very good ... my main concern is that you have to go further with your research, because I'm afraid your current  findings are not enough to create a guideline to identify patients who may avoid the regular UEs based on US-elastography ... But your work is important - still, we don't have many research papers comparing TE and SWE in liver diseases ...

I will wait for your opinions and potential corrections in those matters ...

Best Regards !

Round 2

Reviewer 1 Report

The quality of the manuscript has been improved, but one minor correction needs to be made: the resolution of Figure 3 is low, and the first image was shrunk. 
